# Roles of GacSA and DJ41_1407 in *Acinetobacter baumannii* ATCC 19606

**DOI:** 10.3390/ijms262110620

**Published:** 2025-10-31

**Authors:** Yee-Huan Toh, Meng-Yun Wen, Guang-Huey Lin

**Affiliations:** 1Master Program in Biomedical Sciences, School of Medicine, Tzu Chi University, Hualien 970374, Taiwan; 110333103@gms.tcu.edu.tw; 2Department of Microbiology and Immunology, School of Medicine, Tzu Chi University, Hualien 970374, Taiwan; 106329103@gms.tcu.edu.tw

**Keywords:** two-component system, *Acinetobacter baumannii*, transcriptome analysis, virulence, aminoglycosides resistance, biofilm *formation*, motility

## Abstract

Two-component systems (TCSs) in bacteria are often involved in the global regulation of various physiological activities and behaviours. This study investigated the GacSA TCS and DJ41_1407 transcriptional sensor adjacent to GacA in *Acinetobacter baumannii* ATCC 19606. The relationship between GacS, GacA, and DJ41_1407 and their functions and signal transduction mechanisms are described. *A. baumannii* ATCC 19606 mutants, ∆*gacS*, ∆*gacA*, and ∆*DJ41_1407*, were generated using markerless mutation and cultured in LB medium, then collected for RNA sequencing. It was found that GacS, GacA, and DJ41_1407 regulate a series of genes involved in carbon metabolism. Quantitative reverse transcription PCR (qRT-PCR) results showed that DJ41_1407 and GacA may regulate the expression of *adh4*, *ipdC, iacH*, and *paa*. Phos-tag™ results revealed that GacS plays a more significant role in GacA phosphorylation. GacA regulated colony size and growth conditions in rich medium. Compared to the wild-type strain, the ∆*gacA* and ∆*gacSA* mutants exhibited smaller colony sizes, and mutation of the *gacS*, *gacA*, and *DJ41_1407* genes also reduced bacterial virulence as determined by the *Galleria mellonella* infection assay. GacA also plays a crucial role in modulating antibiotic resistance, and the ∆*gacA*∆*DJ41_1407* mutant demonstrated greater susceptibility to antibiotics. These results highlight the multiple functions regulated by the GacSA global TCS in *A*. *baumannii* ATCC 19606.

## 1. Introduction

*Acinetobacter baumannii*, a Gram-negative bacterium, acts as an opportunistic human pathogen [1] that has garnered attention for its role in causing various severe nosocomial infections, including skin and soft tissue infections, wound infections, urinary tract infections, and secondary meningitis, among others [2,3,4]. Of these, ventilator-associated pneumonia and bloodstream infections are among the most significant, as they are linked to higher mortality rates [5]. *A. baumannii* is classified as an ESKAPE (an acronym of the scientific names for six highly virulent and antibiotic-resistant pathogens, including *Enterococcus faecium*, *Staphylococcus aureus*, *Klebsiella pneumoniae*, *A. baumannii*, *Pseudomonas aeruginosa*, and *Enterobacter* spp.) pathogen due to its high levels of antibiotic resistance [6], and its ability to form biofilms, which enhances its capacity to persist on artificial surfaces within hospital environments for extended periods [7]. *A. baumannii* generally utilises two-component systems (TCSs) to regulate adaptive responses and traits associated with virulence.

TCSs are one of the largest signal transduction pathways in living organisms and play an important role in sensing external signals, regulating response and promoting survival in changing environments. Most TCSs in bacteria can regulate gene expression related to antibiotic resistance [8], virulence, biofilm formation [9], motility [10], and ethanol metabolism [11,12]. Moreover, some of these have been identified as global TCSs, due to their capacity to regulate various physiological activities and behaviours. For example, GacS/GacA in *Pseudomonas fluorescens* MFE01 control antimicrobial activities against phytopathogens and emissions of volatile organic compounds (VOCs) [13]. In *Legionella pneumophila*, PmrAB was found to exert a broad influence on genes encoding eukaryotic-like proteins, the Dot/Icm apparatus, secreted effectors, type II-secreted proteins, regulators of the post-exponential phase, stress response genes, flagellar biosynthesis genes, metabolic genes, and genes of unknown function [14]. It has also been shown to be essential for the intracellular proliferation of *L. pneumophila* within human macrophages and protozoa [14]. Disparities in gene regulation and intracellular growth patterns observed between the *pmrA* and *pmrB* mutants suggest potential crosstalk with other TCSs [14]. Additionally, CovSR (CsrSR) in group B streptococcus has been identified as a global TCS implicated in virulence. Expression profiling analysis revealed that CovSR regulates a substantial array of genes that encode proteins potentially involved in secretion or associated with the cell surface [15]. AirSR is a redox-dependent global regulatory system in *Staphylococcus aureus* that plays essential roles in gene regulation through a redox-active Fe–S cluster under O_2_-limited conditions, and research has shown that an AirSR-mutant strain demonstrated GacSA heightened resistance to H_2_O_2_, vancomycin, norfloxacin, and ciprofloxacin in anaerobic environments [16].

GacSA has been identified as a global regulator in Gram-negative bacteria. In *P. aeruginosa*, GacSA (GacS-PA0928, GacA-PA2586) is one of the crucial TCSs and plays a central role in regulating the expression of virulence factors, secondary metabolites, biofilm formation, and quorum sensing [17,18]. Additionally, GacSA acts as a key switch that determines the transition between acute and chronic infections [19]. In *P. aeruginosa*, the phosphorylation of GacS is regulated by two hybrid sensor kinases, RetS (PA4856) [20] and LadS (PA3974) [21]. RetS directly interacts with GacS, inhibiting its phosphorylation [21,22], while LadS phosphorylates GacS [21]. The phosphorylated GacS then promotes the transcription of two small regulatory RNAs, rgRsmZ (PA3621.1) and rgRsmY (PA0527.1). Both rgRsmZ and rgRsmY function as inhibitors of the negative regulator, the RNA-binding protein RsmA (PA0905). RsmA, in turn, positively regulates genes associated with the Type 3 secretion system, type IV pili formation, and iron homeostasis while repressing quorum sensing, Type 6 secretion, and potentially other transcription factors [23,24,25]. In *K*. *pneumoniae*, the GacSA homologue protein is known as the BarA/UvrY TCS. Studies have demonstrated that inactivation of the *barA* and *uvrY* genes is linked to reduced capsule polysaccharide (CPS) production [26]. Transcriptome and proteome analyses further showed that cefiderocol stress downregulates UvrY transcription and expression levels, indicating that UvrY may play a role in the response to cefiderocol pressure. However, the regulated genes and activation signals associated with the BarA/UvrY TCS system still require further investigation [27].

GacSA is known to control the expression of 674 genes associated with virulence, biofilm formation, pili production, resistance to human serum, motility, metabolism of aromatic compounds, and more in *A. baumannii* ATCC 17978 [28,29]. Unlike other well-characterized TCSs such as AdeRS [30], BaeSR [31], PmrAB [32], and BfmRS [33], GacSA does not exist as a contiguous operon in *A. baumannii* ATCC 17978, and the *GacS* and *GacA* genes are positioned differently on the chromosome. However, the response regulator GacA was found to be transcribed in the same direction as the adjacent sensor kinase of unknown function, namely DJ41_1407 in *A. baumannii* ATCC 19606, indicating potential crosstalk between these two TCSs, as GacA may have co-opted this sensor kinase. Given the differences between strains, the signal transduction and function of GacSA in other strains, such as *A. baumannii* ATCC 19606, remain unclear.

Considering that GacSA functions as an important global regulator in *A. baumannii*, a better understanding of its role in physiological mechanisms could potentially lead to the development of novel strategies that can address issues such as virulence and antibiotic resistance in this critical nosocomial pathogen. In this study, we aim to confirm the roles of GacSA in *A. baumannii* ATCC 19606 and to investigate the relationship between GacA and its adjacent sensor kinase DJ41_1407.

## 2. Results

### 2.1. GacSA Is a TCS in A. baumannii ATCC 19606

The *gacS* gene in *A. baumannii* ATCC 19606 is designated as *DJ41_1085* and annotated as a hybrid sensor histidine kinase/response regulator. The conserved and functional domains of GacS were analysed using CDvist (http://cdvist.joulinelab.org/ (accessed on 28 August 2025); Appendix A), which revealed the presence of a phosphatidate phosphatase domain, a Hamp domain, a histidine kinase domain for phosphorylation, an ATPase domain, a FleQ domain for flagellar regulation, a receiver domain for phosphorylation, and an Hpt domain for phosphate transfer. The potential phosphorylation sites include a histidine residue at position 299 and an aspartic acid residue at position 719. The *gacA* gene in *A. baumannii* ATCC 19606 is designated as *DJ41_1406* and annotated as a response regulator. The conserved and functional domains of GacA were analysed using CDvist (Appendix A), revealing a receiver domain for phosphorylation and a helix-turn-helix domain for DNA binding. The potential phosphorylation site is the aspartic acid residue at position 54.

Phos-tag^TM^ was used to investigate whether GacS phosphorylates GacA, in order to determine if GacS and GacA constitute a TCS in *A. baumannii* ATCC 19606. Wild-type and Δ*gacS* strains were cultured under the same conditions, specifically LB medium or M9 medium supplemented with 5 mM citrate, indole-3-acetic acid (IAA), or tryptophan (Appendix A). The results indicated that when *gacS* was mutated, there was decreased expression of phosphorylated GacA, even across different culture conditions. This finding suggests that GacA is phosphorylated by GacS and that they form a TCS.

### 2.2. GacSA Transcription Analysis

To analyse the functions of GacS and GacA, wild-type, ∆*gacS*, ∆*gacA*, and ∆*gacSA* strains were cultured in LB medium and subsequently sent to Welgene Biotech Co., Ltd. (Taipei, Taiwan) for transcriptome analysis. RNA expression levels of the mutants were compared against wild-type. The fold-change in each mutant relative to the wild-type was presented as a log2 expression ratio (wild-type/mutant). A total of 88, 135, and 169 genes exhibited differences in expression activity compared with the wild-type for the ∆*gacS*, ∆*gacA*, and ∆*gacSA* mutants, respectively. Among the regulated genes in each group, 44 genes were found to be co-regulated by GacS, GacA, and GacSA, indicating that the combination of GacS and GacA plays a significant role in key regulatory processes. Additionally, 67 genes were co-regulated solely by GacA and GacSA, suggesting that GacA may exclusively regulate the expression of these genes in response to phosphorylation by other sensor kinases (Figure 1A; Appendix A).

To gain a deeper understanding of GacSA regulation, genes with a log2 expression ratio (wild-type/∆gacSA) greater than 1.0 and less than −1.0 were analysed and classified through gene ontology and KEGG pathway classification. In the gene ontology classification, the genes regulated by GacSA were found to be involved in molecular functions (59.2%), cellular components (23.2%), and biological processes (17.6%) (Figure 1B). Most of the genes regulated by GacSA were associated with molecular functions; for instance, 16 genes were linked to oxidoreductase activity, and 14 genes were related to catalytic activity. KEGG pathway classification revealed that, of the genes regulated by GacSA, 35.8%, 30.6%, 23.3%, 6.7%, and 3.6% were respectively involved in carbon metabolism, other metabolic processes, amino acid metabolism, fatty acid metabolism, and nucleic acid metabolism. GacSA primarily regulated genes associated with carbon metabolism and amino acid metabolism; specifically, 18 genes were related to benzoate degradation, while 13 genes were linked to phenylalanine metabolism (Figure 1C).

### 2.3. GacA and DJ41_1407 Are Expressed in the Same Transcript

Amino acid identity analysis revealed that DJ41_1407 and GacA can be found in most *Acinetobacter* spp. Genome structure analysis of the neighbour regions of the *gacA* and *DJ41_1407* gene clusters revealed that this gene cluster is also found in *Klebsiella pneumonia* and *Prolinoborus fasciculus*, with high amino acid identity among homologues. The gene organization and gene orientation are also identical between different strains, which shows that this gene cluster is highly conserved among bacterial species (Figure 2).

TCS genes in bacterial chromosomes are often located adjacent to one another. In the *A. baumannii* ATCC 19606 genome, the *gacA* and *DJ41_1407* genes are transcribed in the same direction. Their intergenic region is only 55 base pairs long, suggesting that they may constitute a TCS. RNA was extracted from the wild-type strain, and reverse transcription PCR (RT-PCR) was performed to analyze the transcripts of these genes. The cDNA was used to amplify the intergenic region between *gacA* and *DJ41_1407*. If *gacA* and *DJ41_1407* are expressed from the same transcript, a product should be successfully amplified; otherwise, no product would be obtained. The results demonstrated that the intergenic region between *gacA* and *DJ41_1407* was successfully amplified (Figure 3A, lane 3), indicating that these two genes are transcribed from the same mRNA.

### 2.4. GacA and DJ41_1407 Constitute a TCS

DJ41_1407 has been identified as a sensor kinase. The original amino acid sequences of DJ41_1407 were acquired from GenBank (https://www.ncbi.nlm.nih.gov/genbank/ (accessed on 28 August 2025)), and the conserved and functional domains were analysed using Cdvist (Appendix A), which revealed a per-Arnt sim domain for signal sensing, a histidine kinase domain for phosphorylation, and an ATPase domain. The potential phosphorylation site is the histidine residue at position 308. To investigate the interaction between GacA and DJ41_1407 in *A. baumannii* ATCC 19606, Phos-tag^TM^ was used to determine whether DJ41_1407 phosphorylates GacA. The wild-type and Δ*DJ41_1407* strains were cultured under the same conditions (Figure 3B). The results indicated that when *DJ41_1407* was mutated, a decrease in phosphorylated GacA was observed in cultures in M9 medium with 5 mM citrate, IAA, or pyruvate. This suggests that GacA is phosphorylated by DJ41_1407 and that they function as a TCS. However, the level of GacA phosphorylation by DJ41_1407 was lower compared to that of GacS on GacA (Appendix A). In summary, although GacS plays a more significant role in GacA phosphorylation, DJ41_1407 can also phosphorylate GacA under culture conditions (Figure 3B).

### 2.5. Transcriptome Analysis of GacA-Regulated Genes

To analyse the function of GacS and GacA, wild-type and ∆*DJ41_1407*, ∆*gacA*, and ∆*DJ41_1407*∆*gacA* mutant strains were cultured in LB medium and were sent to Welgene Biotech Co., Ltd. for transcriptome analysis. A total of 47, 135, and 185 genes respectively showed differences in expression activity compared to wild-type for the ∆*DJ41_1407*, ∆*gacA*, and ∆*DJ41_1407*∆*gacA* mutants (Figure 4A; Appendix A). Considering that the gene ontology classification results showed that GacA regulates many genes that may be associated with bacterial metabolism, genes and gene clusters with expression differences greater than 2-fold (log2 greater than 1.0 or less than −1.0) were independently analysed to investigate this further. Analysis revealed a 5-fold decrease in gene expression for *DJ14_136* in Δ*gacA* (Figure 4B), for which the translated protein is alcohol dehydrogenase (Adh4, *DJ41_136*). Previous studies have shown it to be primarily involved in the alcohol metabolism of *A. baumannii*, converting ethanol to aldehyde [34]. *DJ14_133* showed a 3.5-fold decrease in gene expression in Δ*gacA*, and its translated protein is annotated as aldehyde dehydrogenase (ALDH), responsible for converting aldehyde to less toxic acetate. DJ41_1668, also annotated as an ALDH, exhibited similar down-regulation in Δ*gacA* (Figure 4B). In the Δ*gacA* transcriptome analysis, *DJ41_1670* showed a 6-fold decrease in gene expression, and its translated protein, indole-3-pyruvate decarboxylase (IpdC), has been demonstrated to participate in the production of the plant hormone IAA in *A. baumannii* [35]. Genes *DJ41_358* and *DJ41_359*, with respectively 2.2- and 3.3-fold decreased expression in Δ*gacA*, encode IacH (indole-3-acetic acid catabolism) and IacA, both previously linked to the breakdown of the plant hormone IAA [24,25] (Figure 4B).

Furthermore, Δ*gacA* transcriptome analysis revealed that the gene cluster *DJ41_2041*-*DJ41_2054* exhibited a greater than 2-fold decrease in gene expression in Δ*gacA* (Figure 4C). The proteins translated from *DJ41_2041-DJ41_2054* were previously shown to be involved in the degradation of phenylacetic acid (PAA), indicating that GacA regulates the metabolisation of alcohols into acetic acid. Acetic acid is then converted to IAA, which, through the PAA degradation pathway, is further degraded into acetyl-CoA and succinyl-CoA, ultimately entering the citric acid cycle for energy production (Appendix A). In addition, three gene clusters, *DJ41_332*-*DJ4_338*, *DJ41_2892*-*DJ41_2896*, and *DJ41_3071*-*DJ41_3075*, exhibited more than a 2-fold increase (log2 of 1.0) in gene expression in Δ*gacA* (Figure 4D). The functions of the translated proteins from these clusters were analysed using the KEGG enzyme database (https://www.genome.jp/kegg/annotation/enzyme.html (accessed on 28 August 2025)). DJ41_332-DJ4_338 is involved in benzoate metabolism, DJ41_2892-DJ41_2896 is associated with arginine and proline metabolism, and DJ41_3071-DJ41_3075 is involved in alanine, aspartate, and glutamate metabolism. These clusters convert amino acids into succinate and oxaloacetate, which also enter the citric acid cycle for energy production (Appendix A).

### 2.6. GacA Regulates Genes Involved in Alcohol, IAA, and Phenylacetic Acid Metabolism

To validate the transcriptome analysis results, real-time quantitative polymerase chain reaction (qRT-PCR) was conducted to measure gene expression levels of *adh4*, *ipdC*, *iacH*, and *paa* in each strain. The results showed a significant reduction in *adh4* expression in the Δ*DJ41_1407* strain, suggesting that DJ41_1407 modulates the expression of the *adh4* gene. In both Δ*gacA* and Δ*DJ41_1407*Δ*gacA* strains, the expression of the *adh4* gene is nearly absent, implying that in *A*. *baumannii*, transcription of the *adh4* gene is primarily regulated by GacA (Figure 5A). In comparison to the wild-type, *ipdC* expression is nearly absent in both Δ*gacA* and Δ*DJ41_1407*Δ*gacA* strains (Figure 5B). The results for *iacH* gene expression also indicate that in the Δ*gacA* and Δ*DJ41_1407*Δ*gacA* strains, *iacH* gene expression is almost negligible (Figure 5C). An examination of the expression levels of the *paa* gene revealed a reduction in *paa* expression in the Δ*DJ41_1407* strain, while in the Δ*gacA* and Δ*DJ41_1407*Δ*gacA* strains, *paa* gene expression is almost absent (Figure 5D). The qRT-PCR results are consistent with the transcriptomic analysis results, indicating that DJ41_1407/GacA not only participates in amino acid metabolism but also regulates bacterial metabolism in alcohol, IAA and phenylacetic acid. This underscores the crucial role of DJ41_1407/GacA in bacterial energy acquisition.

### 2.7. Possible Binding Region of GacA

The promoter regions of gene clusters regulated by GacSA were analysed using MEME to identify potential GacA binding boxes. The GacA binding box is within a 15 bp region, with a 5′ CAAAWWAAAGTTGCA 3′ sequence on the positive strand (Figure 6A). These sequences were also consistent with those in the upstream region of genes exhibiting opposite transcriptional orientations (Figure 6B). Previous studies have shown that the regulator of a TCS binds to its promoter to self-regulate its expression, a phenomenon also observed in GacA. The GacA binding box was identified in the upstream region from −103 to −117 of DJ41_1406, with the 5′ CAAAATAAGCTTGCA 3′ sequence. This finding suggests that GacA regulates its own expression, a characteristic commonly observed in most bacterial TCS (Figure 6C).

### 2.8. GacA Regulates Bacterial Growth

To understand the biological role of GacA and its corresponding sensor, colonies of *A. baumannii* ATCC 19606 and its mutants were observed on LB agar plates using a stereoscopic microscope. The results indicated that the colony sizes of the ∆*gacA* and ∆*gacSA* mutants were significantly smaller than those of the other strains (Figure 7A). ImageJ (https://imagej.net/ij/ (accessed on 28 August 2025)) was utilised to analyse colony diameter and revealed that the colony sizes of ∆*gacA* were 1.28-fold smaller than those of all other strains, except for ∆*gacSA*. Additionally, ∆*gacSA* exhibited colony sizes that were 1.21-fold smaller than those of *A. baumannii* ATCC 19606, ∆*gacS*, and ∆*DJ41_1407*. No significant differences were observed among the wild type, ∆*gacS*, ∆*DJ41_1407*, ∆*gacS*∆*DJ41_1407*, and ∆*gacA*∆*DJ41_1407* strains (Figure 7B). The growth curves of the wild-type and mutant strains were assessed in LB medium over 12 h. The results indicated a significantly reduced optical density (OD) in the ∆*gacA* and ∆*gacA*∆*DJ41_1407* mutants (Figure 7C). These results suggest that GacA can play a role in colony size and growth conditions in rich medium for *A. baumannii* ATCC 19606.

### 2.9. GacSA Regulates Virulence

To determine whether GacS, GacA, and DJ41_1407 can influence the virulence of *A. baumannii*, we infected *Galleria mellonella* larvae with *A. baumannii* ATCC 19606 and its mutant strains. The survival rates of the larvae were monitored over 96 h. Results indicated that larvae injected with Phosphate-Buffered Saline (PBS) survived throughout the 96 h, demonstrating health and no mortality. In contrast, larvae injected with the wild-type strain exhibited a survival rate of only 20% after 24 h, indicating high bacterial virulence. Conversely, larvae infected with the Δ*gacA* and Δ*gacSA* strains showed a survival rate of 80%, while those infected with Δ*gacS* had a survival rate of 70%. This suggests that bacterial virulence was reduced after the *gacS* and *gacA* genes were mutated. Additionally, larvae infected with the Δ*gacA*Δ*DJ41_1407* strain demonstrated a 60% survival rate after 96 h, indicating that bacterial virulence was also reduced when both the *gacA* and *DJ41_1407* genes were mutated. These results suggest that the *gacS*, *gacA*, and *DJ41_1407* genes play a crucial role in regulating the expression of bacterial virulence genes (Figure 7D).

### 2.10. GacA Regulates Antibiotic Resistance

Previous studies have demonstrated that TCSs can significantly influence bacterial resistance to antibiotics (8). Therefore, we aimed to determine whether GacS, GacA, and DJ41_1407 might also affect the antibiotic resistance of *A. baumannii* ATCC 19606. Wild-type and mutant strains were cultured in media containing varying concentrations of apramycin, gentamicin, kanamycin, chloramphenicol, polymyxin B, and colistin, to assess the minimum inhibitory concentration (MIC) of these antibiotics for each strain (Table 1). For the ∆*gacA*∆*DJ41_1407* strain, there was a 4-fold decrease in the MIC for apramycin and a 2-fold reduction in the MICs for gentamicin, kanamycin, and chloramphenicol, compared to the wild-type strain. The MICs of all tested antibiotics for the ∆*gacS*, ∆*DJ41_1407*, and ∆*gacS*∆*DJ41_1407* strains remained unchanged from those of the wild-type (Table 1). These findings indicate that GacA plays a crucial role in modulating the antibiotic resistance of *A. baumannii* ATCC 19606.

## 3. Discussion

The GacSA TCS has been identified as a global regulator in *A. baumannii* ATCC17978, controlling the expression of 674 genes associated with virulence, biofilm formation, pili production, resistance against human serum, motility, and metabolism of aromatic compounds [28,29]. In this study, GacSA was found to regulate virulence in *A. baumannii* ATCC 19606, exhibiting certain similarities with GacSA in *A. baumannii* ATCC 17978. In both *A. baumannii* ATCC 19606 and *A. baumannii* ATCC 17978, GacSA was observed to affect the expression of genes involved in virulence and biofilm formation, as well as the metabolism of aromatic compounds, such as the catabolic pathway of aromatic compounds known as the PAA pathway, which was also investigated in this study.

In this study, we discovered that GacA was phosphorylated by both GacS (Appendix A) and DJ41_1407 (Figure 3B), indicating crosstalk between GacSA, DJ41_1407, and GacA. This phenomenon is also observed in other bacterial species. In *Serratia marcescens*, QseBC is involved in quorum sensing, while RssAB plays a role in swarming and biofilm formation [36]. QseC can dephosphorylate RssB ∼ P, deactivating RssAB signaling and facilitating bacterial surface migration initiation during swarming development. These findings highlight the crosstalk between two TCSs, which cooperatively regulate flagellar biosynthesis in a stage-specific manner during swarming development [36]. Another unique crosstalk system between TCSs has been observed in *E. coli*, involving the HprR and CusR response regulators [37,38]. HprSR is involved in the stress response to hydrogen peroxide, while CusSR is responsible for the response to Cu(II) [37,38]. Interestingly, HprR and CusR recognize the same binding sequence, suggesting that they can regulate the same target genes, but cannot bind to these targets simultaneously. Both regulators recognize and transcribe the *hiuH* and *cusC* promoters, albeit with different efficiencies, indicating a collaborative mechanism [37,38]. However, as protein concentrations increase, HprR and CusR compete to transcribe common targets, demonstrating a competitive interplay [37,38]. Further research is needed to ascertain whether GacSA and DJ41_1407 have similarly competitive crosstalk mechanisms in place.

We further found that GacSA in *A. baumannii* enhanced resistance against apramycin, gentamicin, and kanamycin (Table 1). In comparison, GacSA in *P. aeruginosa* has also been implicated in antibiotic resistance against three different antibiotic families, tobramycin, ciprofloxacin, and tetracycline [39], likely through the RsmA/rgRsmZ pathway. The BfiRS system potentially collaborates with GacSA and contributes to the regulation loop involving Rsm, thereby exerting control over biofilm formation [40]. Compared to GacSA in *P. aeruginosa*, fewer functions of GacSA in *A. baumannii* ATCC 19606 have been identified for now, but key similarities can be noted and may point the way for future research.

This study observed that virulence of *A. baumannii* ATCC 19606 against *G. mellonella* larvae was significantly reduced after the deletion of *gacS* and/or *gacA* genes (Figure 7D). It has been previously reported in *A. baumannii* ATCC 17978 [29] that mutation of the *gacS* gene results in avirulence of the mutated strains against *Candida albicans* and mice. The study further noted that virulence was restored by complementation [29]. Previous research showed that dissemination from infection sites to other organs in murine models was much reduced with *gacS* deletion mutant strains compared to wild type, and the mutant strains were also more susceptible to the killing effects of human serum [29]. This has been linked to the PAA pathway, as the *paa* operon is completely repressed in *gacS* deletion mutant strains, thus preventing them from utilising L-phenylalanine as a carbon source [29]. This study also observed that deletion of the *gacS* and/or *gacA* genes downregulates the expression of key genes involved in alcohol and amino acid metabolism, which can limit energy sources available to the mutant strains (Figure 4; Appendix A). Moreover, ∆*gacSA* mutant strains exhibited smaller colony sizes (Figure 7B). These results suggest that disruption of the GacSA TCS has profound effects on bacterial metabolism, growth, dissemination, and virulence, and may have implications for the development of novel antibacterial strategies.

Together, the findings in this study show that GacSA as well as DJ41_1407/GacA are TCSs and further indicate crosstalk between these two TCSs. GacA can be phosphorylated by both GacS and DJ41_1407, with GacS playing a more significant role in GacA phosphorylation than DJ41_1407. Together, GacS, GacA, and DJ41_1407 were found to regulate key functions and genes involved in benzoate degradation, phenylalanine metabolism, virulence, and antibiotic resistance of *A. baumannii* ATCC 19606. These findings shed light on the GacSA TCS in *A. baumannii* ATCC 19606 and offer important insights regarding the regulation of energy production, virulence, and resistance.

## 4. Materials and Methods

### 4.1. Bacterial Strains, Plasmids, Culture Media, and Markerless Mutation

The bacterial strains used in this study are detailed in Table 2. The plasmids and primers employed in constructing the mutants are respectively listed in Table 3 and Table 4. LB medium (0.5% yeast extract, 1% tryptone, and 1% NaCl [41]; Becton Dickinson (BD), Franklin Lakes, NJ, USA) and agar (BD) were utilised to culture the bacterial strains statically or with shaking at 200 rpm at 37 °C. PCR-amplified 1 kb fragments of the upstream and downstream regions of *GacS*, *GacA*, and *DJ41_1407* genes were inserted into the plasmid pK18mobsacB and transformed into the donor strain *E. coli* S17-1λπ. The recombinant plasmid containing the donor was then introduced into *A. baumannii* through conjugation. The transconjugants were cultured on LB plates supplemented with ampicillin and kanamycin to select for *A. baumannii* that harboured the integrated recombinant pK18mobsacB, indicating the occurrence of the first homologous recombination. Colony PCR confirmed successful recombination. The positive colony was subsequently cultured in LB with 20% sucrose, resulting in a second homologous recombination with a 50% probability of deleting the desired gene fragment [42].

### 4.2. Transcriptome Sequencing and Analysis

Bacterial strains were cultured in LB medium containing 50 μg/mL of ampicillin at 37 °C for 12 to 14 h and then subcultured with an initial optical density (OD) at 600 nm (OD_600_) of 0.1 for 3 h. After this incubation period, the bacterial solution was aliquoted to achieve an OD_600_ of 0.6 in each microcentrifuge tube. Subsequently, 1/10 of the volume of acid phenol [5% acid phenol, 95% ethanol] was added to the microcentrifuge tubes and mixed thoroughly to fix the samples. The microcentrifuge tubes were centrifuged at 4 °C and 17,000× *g* for 10 min. Following centrifugation, the supernatant was carefully removed, and the pellet was resuspended in 1 mL Thermo Fisher TRIzol™ Reagent (Invitrogen, Waltham, MA, USA). After resuspension, the samples were stored at −80 °C and were sent to Welgene Biotech Co., Ltd. for transcriptome analysis [8]. RNA sequence reads were generated using Illumina HiSeq 2000 (San Diego, CA, USA) and then underwent selection to remove reads containing adaptors, reads containing over 10% of unknown sequences, and reads of low quality (defined as more than half of all bases having a quality score < 5). The curated set of reads was then mapped to the *A. baumannii* ATCC 19606 genome published on GenBank (accession numbers: SRX3312085 and SRX3312086) using SOAP2 software (https://bioinformaticshome.com/db/tool/SOAP2 (accessed on 28 August 2025)) [44]. We subsequently calculated gene expression levels as reads per kilobase of genes per million reads (RPKMs) [45,46], and false discovery rates (FDRs) [47] and the RPKM ratios of two samples were utilized to discern differentially expressed genes (DEGs). We considered DEGs to be genes with FDR ≤ 0.01 and an absolute log2 value ratio of greater than 1.

### 4.3. RNA Extraction and Reverse Transcription (RT)

Bacterial strains were cultured at 37 °C with agitation overnight and then subcultured in 50 mL of LB medium for 3 h after OD_600_ reached 0.3. A total of 0.6 of OD_600_ samples were then collected and mixed with 0.1 volume of fixing solution (5% acid phenol, 95% ethanol). After centrifugation at 17,000× *g* at 4 °C, the cell pellets were stored at −80 °C for RNA extraction. Cell pellets were thawed on ice and resuspended in 1 mL of NucleoZOL (MACHEREY-NAGEL, Düren, Germany), then mixed thoroughly with 400 μL of diethyl pyrocarbonate (DEPC)-treated H_2_O, and incubated at room temperature for 15 min. The supernatant was recovered after centrifugation at 17,000× *g* at 4 °C for 20 min and then mixed with 5 μL of 100% 4-bromoanisole and incubated at room temperature for 10 min. Excess protein was then removed by centrifugation at 17,000× *g* at 4 °C for 20 min. The resulting RNA suspension was mixed with an equal volume of isopropanol for 15 min to induce RNA precipitation. The derived RNA pellet was washed twice with ice-cold 75% ethanol and resuspended in 30 μL of DEPC-treated H_2_O for analysis [34].

A total of 2 μg of RNA was used to prepare cDNA. The RT-PCR mixture contained 10× reaction buffer, 200 U of MMLV high-performance reverse transcriptase (Epicentre, Madison, WI, USA), 100 mM of dithiothreitol (DTT), 2.5 mM dNTP, and 1 nM of hexamer. The reaction was conducted in a Biometra TADVANCED Thermal Cycler (Analytik Jena, Jena, Germany). The gyrase gene served as an internal control, and the other gene-specific primers used to determine the presence and expression levels of the respective genes are listed in Table 4 [35].

### 4.4. Phos-Tag^tm^ Acrylamide Gel Electrophoresis

Wild-type, Δ*gacS*, and Δ*DJ41_1407* strains were cultured in LB broth with shaking at 37 °C overnight. Bacteria were subcultured in M9 medium containing 5 mM citrate, IAA, or tryptophan. Bacteria were collected after OD_600_ reached 0.6 and then centrifuged and resuspended in 130 μL of 1 M formic acid, 54 μL of sample buffer [130 mM Tris-Cl (pH 6.8), 6% SDS, 15% β-mercaptonethanol, 3% glycerol, 15% bromophenol blue], and 24 μL of 5 N NaOH. Phos-tag^TM^ containing 12% sodium dodecyl sulfate (SDS)-acrylamide resolving gel was prepared with 50 μL of 50 mM Phos-tag^TM^, 1.25 mL of 1.5 M Tris-Cl (pH 8.8), 2.12 mL of 30% acrylamide solution, 50 μL of 10 mM MnCl_2_, 50 μL of 10% SDS, 1.25 mL of ddH_2_O, 25 μL of 10% ammonium persulfate (APS), and 5 μL of N, N,N’,N’-tetramethylethylenediamine (TEMED). A 6% staking gel was prepared with 1 mL of 0.5 M Tris-Cl (pH 6.8), 800 μL of 30% acrylamide solution, 40 μL of 10% SDS, 2.12 mL of ddH_2_O, 40 μL of 10% APS, and 4 μL of TEMED. Electrophoresis of prepared samples was conducted at 120 V at room temperature for 90 min [48].

### 4.5. G. mellonella Larvae Infection Assay

Bacterial strains were cultured in LB medium at 37 °C for 12–14 h. The overnight bacterial solution was washed twice with PBS buffer (0.14 M NaCl, 2.7 mM KCl, 8.1 mM Na_2_HPO_4_, 1.5 mM KH_2_PO_4_) to remove LB medium. The bacterial solution was then diluted to 5 × 10^8^ CFU/mL. Larvae were infected with only PBS and heat-killed wild-type bacteria as the control group (n = 10). Heat-killed wild-type bacteria were prepared by heating the bacterial solution at 100 °C for 5 min. Using a Hamilton syringe [Hamilton^®^ syringe, 700 series, 701N, Reno, NV, USA], 10 μL of diluted bacterial solutions was injected into *G. mellonella* larvae through the last left pro-leg of the abdominal region. The final bacterial concentration was 5 × 10^6^ CFU/larva. Infected larvae were incubated at 37 °C. Survival (alive/dead) and melanisation scores were measured every 24 h for 96 h [49].

### 4.6. MIC Test

The MIC test protocol is based on a modified version of the broth dilution method established by the Clinical and Laboratory Standards Institute (CLSI) and European Committee on Antimicrobial Susceptibility Testing (EUCAST) guidelines [50], and the concentrations of the antibiotics used were based on results derived from previous studies [8,51]. Bacteria were incubated in a shaking incubator at 37 °C for 12–16 h. A 96-well culture plate was prepared, with 270 μL of LB medium added to the first well and 135 μL of LB medium added to each subsequent well. Using kanamycin (Sigma-Aldrich, St. Louis, MO, USA) as an example, a final concentration of 200 μg/mL kanamycin was added to the first well. Half of the volume (135 μL) was transferred to the second well (½ dilution) thoroughly, and then ½ was serially diluted to each subsequent well. After dilution, 135 μL of LB medium was added to each well. The first well had a final concentration of 100 μg/mL kanamycin. Bacterial strains were respectively treated with the following antibiotics (Sigma-Aldrich): apramycin, gentamycin, kanamycin, tetracycline, polymyxin B, and colistin. Each desired antibiotic experiment group was serially performed with six replicates. Overnight bacterial solution was adjusted to OD_600_ = 1.0, and 30 μL was added into each well to make the initial OD_600_ 0.1. After incubating at 37 °C for 12 h, the OD_600_ value was measured to determine the MIC of the antibiotic. The MIC was defined as the antibiotic concentration that was twice the concentration at which there was no bacterial growth [8].

## Figures and Tables

**Figure 1 ijms-26-10620-f001:**
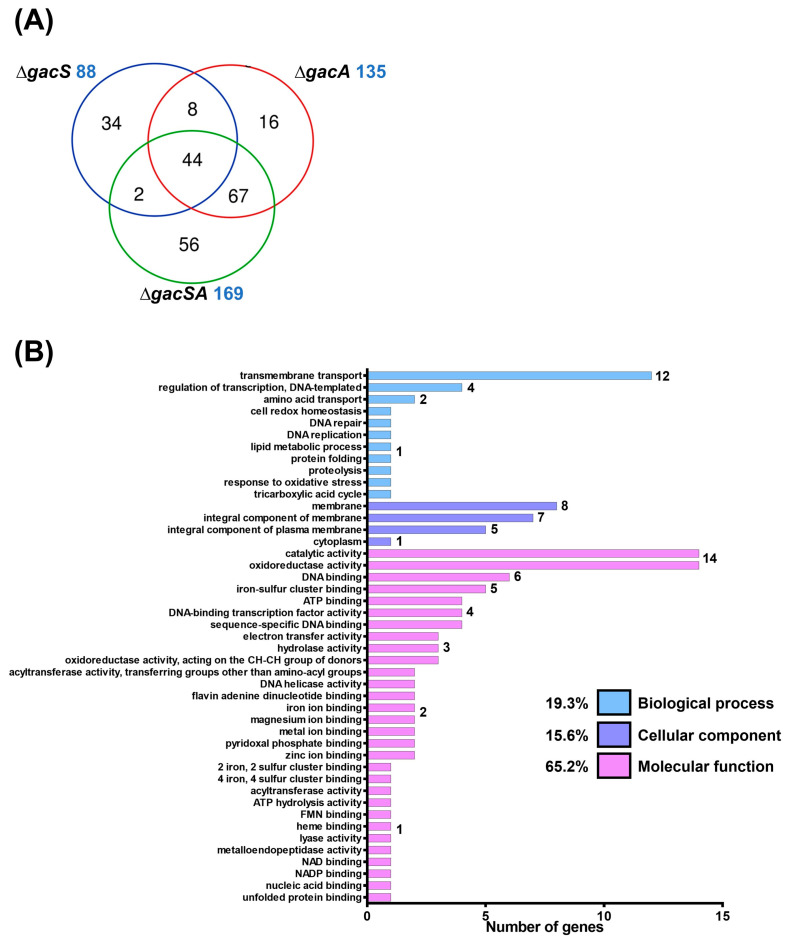
Transcriptome analysis of the GacSA regulon. (**A**) Comparison between the total number of genes regulated by GacS (blue), GacA (red) and GacSA (green). Genes with log2 expression ratio (wild-type/mutant) > 1.0 or < −1.0 in ∆*gacS*, ∆*gacA* and ∆*gacSA* were compared. A total of 88, 135 and 169 genes respectively showed difference in expression activity for the ∆*gacS*, ∆*gacA* and ∆*gacSA* mutants, compared with wild type. (**B**) Metabolic classification of genes regulated by GacSA through gene ontology classification. The indicated gene function groups were divided into three categories based on their roles in biological processes (blue), cellular components (purple), and molecular functions (pink). The number of genes found in these gene function groups ranged from 1 to 17, and these numbers are labelled on the right of the graph bar. (**C**) Metabolic classification of genes regulated by GacSA through KEGG classification. The indicated gene function groups were divided into five categories based on their roles in carbon metabolism (yellow), fatty acid metabolism (green), amino acid metabolism (blue), nucleic acid metabolism (purple), and others (pink). The number of genes found in these gene function groups ranged from 1 to 18, and these numbers are labelled on the right of the graph bar.

**Figure 2 ijms-26-10620-f002:**
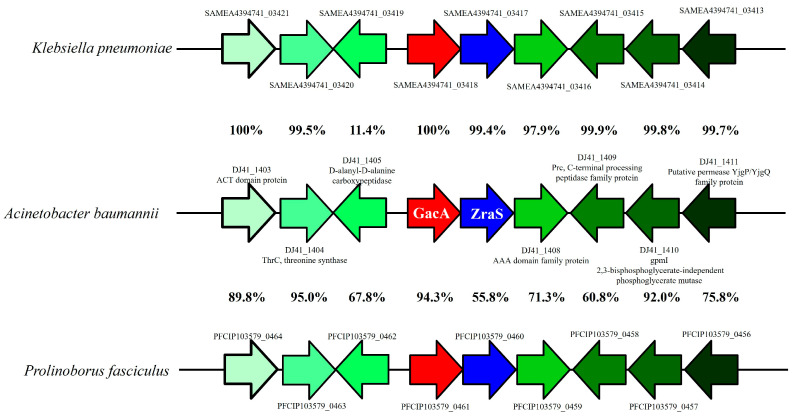
Comparative genomic analysis of the *gacA* and *DJ41_1407* gene clusters and neighbouring regions. The percentages between two gene clusters represent the amino acid similarity between homologues of two different species.

**Figure 3 ijms-26-10620-f003:**
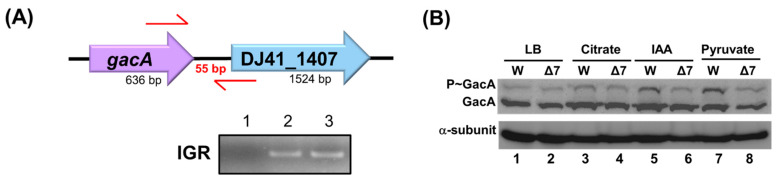
*gacA* and *DJ41_1407* expression in the same transcript. (**A**) Analysis of polymerase chain reaction (PCR) products from the *gacA* and *DJ41_1407* intergenic region. 1: Negative control, 2: *A. baumannii* chromosome as template (positive control), 3: *A. baumannii* complementary DNA (cDNA) as template. IGR: *gacA* and *DJ41_1407* intergenic region amplification. (**B**) Analysing the intensity of phosphorylated GacA protein with Phos-tag^TM^. W: *A. baumannii* ATCC 19606; Δ7: *A. baumannii* ATCC 19606 ΔDJ41_1407; LB: strains cultured in LB medium; Citrate: strains cultured in M9 medium with 5 mM citrate; IAA: 5 mM IAA added; Pyruvate: 5 mM pyruvate added.

**Figure 4 ijms-26-10620-f004:**
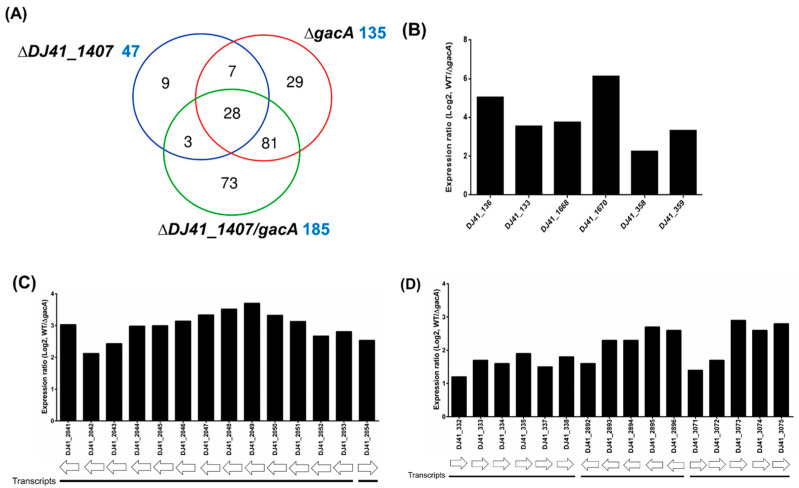
Transcriptome analysis of the DJ41_1407 and GacA regulon. (**A**) Comparison between the total number of genes regulated by DJ41_1407 (blue), GacA (red) and DJ41_1407/GacA (green). Genes with log2 expression ratio (wild-type/mutant) > 1.0 or < −1.0 in ∆*DJ41_1407*, ∆*gacA,* and ∆*DJ41_1407/gacA* were compared. A total of 47, 135 and 185 genes, respectively, showed differences in expression activity for the ∆*DJ41_1407*, ∆*gacA,* and ∆*DJ41_1407*/*gacA* mutant strains compared with the wild-type. (**B**) Gene clusters regulated by GacA with an expression fold-change greater than 2. (**C**,**D**) Gene clusters regulated by GacA are involved in benzoate degradation and the phenylalanine pathway. Gene cluster (**C**) *DJ41_2041*-*DJ41_2054* was found to be related to phenylalanine metabolism through KEGG analysis. Gene clusters (**D**) *DJ41_332*-*DJ4_338*, *DJ41_2892*-*DJ41_2896* and *DJ41_3071*-*DJ41_3075* were found to be related to benzoate degradation and amino acid metabolism.

**Figure 5 ijms-26-10620-f005:**
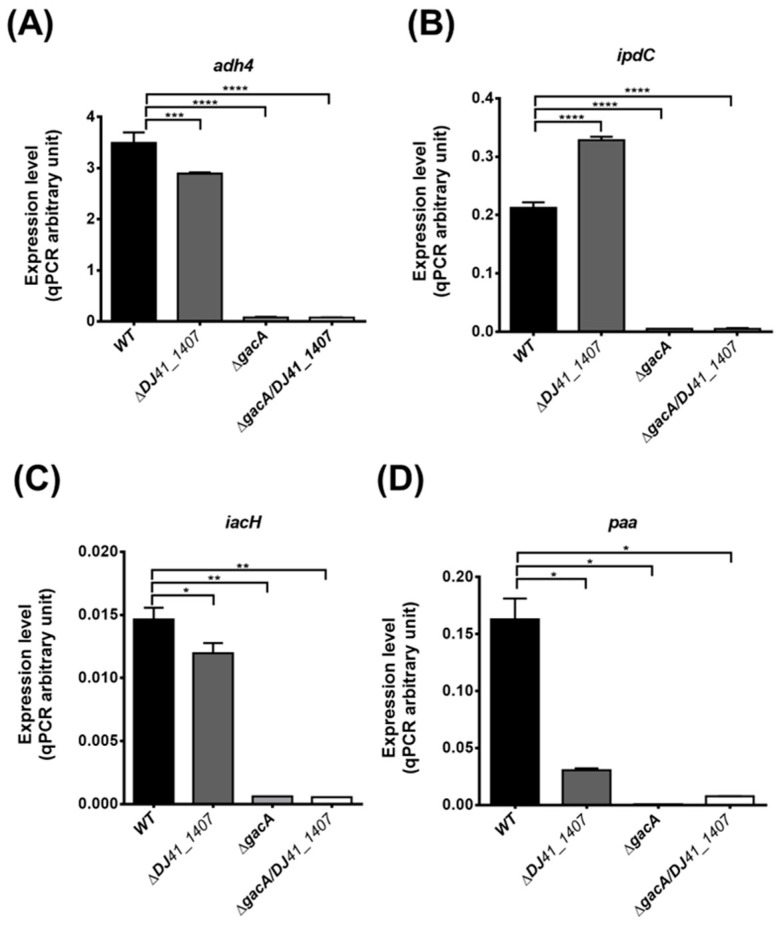
Real-time quantitative PCR analysis of gene expression in wild-type and ∆*DJ41_1407*, ∆*gacA* and ∆*DJ41_1407*/*gacA* mutants. Expression differences in (**A**) *adh4*, (**B**) *ipdC*, (**C**) *iacH* and (**D**) *paa* genes. One-way analysis of variance (ANOVA) was adopted to evaluate the significance of differences. * *p* value < 0.05; ** *p* value < 0.01; *** *p* value < 0.001; **** *p* value < 0.0001.

**Figure 6 ijms-26-10620-f006:**
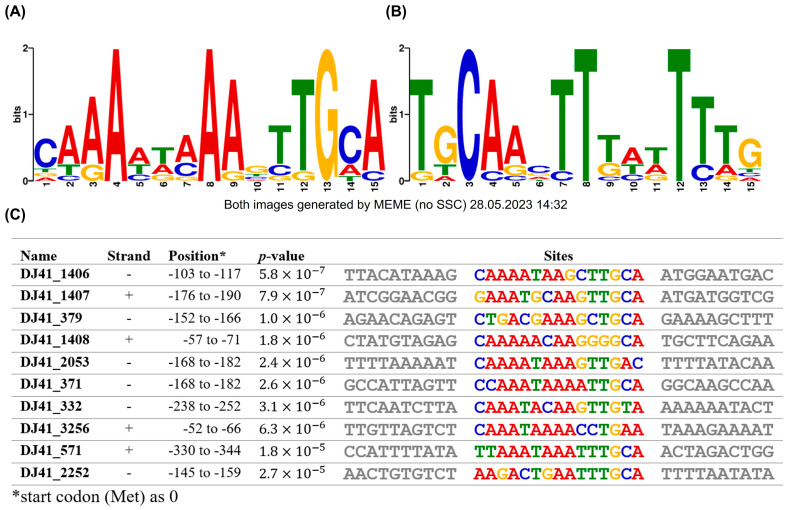
Putative GacA binding box analysis by MEME Suite. Conservation of the GacA binding box in (**A**) positive strand and (**B**) negative strand. (**C**) Exact sequence of the GacA binding box in the promoter region of different genes. The positions represent the location of the conserved sequence upstream of analysed genes, while the first base pair of the start codon was considered as +1. The highly conserved sequence location is likely to be the binding box of GacA.

**Figure 7 ijms-26-10620-f007:**
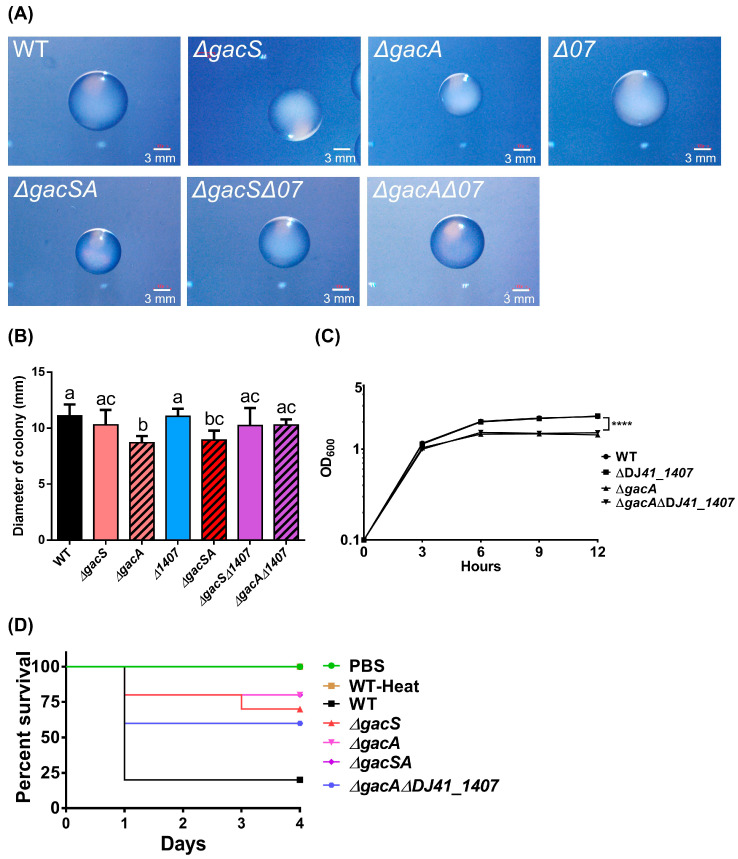
Biological properties of *A. baumannii* ATCC 19606 wild-type and mutant strains. (**A**) Images of colonies were taken at 12 h post-cultivation. The length of the white scale bar represents 3 mm. (**B**) ImageJ was used to analyse colony diameter, using the scale bar as a standard. Each strain was measured in 10 colony replicates. A graph bar with the same alphabet indicates that there were no significant differences between each bar (*p* < 0.05). (**C**) The growth curve of different strains cultured in LB medium, including wild-type (WT), Δ*DJ41_1407*, Δ*gacA* and Δ*gacA*Δ*DJ41_1407*. One-way ANOVA was adopted to evaluate the significance of differences. **** *p* value < 0.0001. (**D**) Survival rate of *G. mellonella* larvae after bacterial infection. The *x*-axis indicates the time from infection of *G. mellonella* larvae by different bacterial strains, while the *y*-axis represents the survival rate of *G. mellonella* larvae in each group. *G. mellonella* larvae were infected with heat-treated wild-type (WT-Heat), wild-type (WT), Δ*gacS*, Δ*gacA*, Δ*gacSA* and Δ*gacA*Δ*DJ41_1407* strains at 5 × 10^6^ CFU/larvae for 4 days (n = 10), or injected with Phosphate-Buffered Saline (PBS).

**Table 1 ijms-26-10620-t001:** Minimum inhibitory concentrations (MICs) of wild-type and mutant bacterial strains against different antibiotics.

STRAIN	APR	GEN	KAN	CHL	PMB	CST
Wild-type	50	25	6.25	30	3.125	2.5
*∆gacS*	50	25	6.25	30	3.125	2.5
*∆gacA*	** 25 **	** 6.25 **	** 3.125 **	** 15 **	** 1.78 **	2.5
*∆gacSA*	** 25 **	** 6.25 **	** 3.125 **	** 15 **	3.125	2.5
*∆DJ41_1407*	50	25	6.25	30	3.125	2.5
*∆ gacS DJ41_1407*	50	25	6.25	30	3.125	2.5
*∆ gacA DJ41_1407*	** 12.5 **	** 6.25 **	** 3.125 **	** 15 **	3.125	2.5

APR: Apramycin, GEN: gentamicin, KAN: kanamycin, CHL: chloramphenicol, PMB: polymyxin B, CST: colistin. Bold underline: 2-fold lower MIC in comparison to wild-type. Bold grey highlight: 4-fold lower MIC in comparison to wild-type. Antibiotic units are in μg/mL.

**Table 2 ijms-26-10620-t002:** Bacterial strains used in this study.

Bacteria	Description	References or Sources
*E. coli*		
DH5α	F^-^, *supE44*, *hsdR17*, *recA1*, *gyrA96*, *endA1*, *thi-1*, *relA1*, *deoR*, λ	ATCC53868
DH5α/pK18_∆*gacA*	Kan ^r^, DH5α containing pK18_∆*gacA*	This study
DH5α/pK18_∆*gacS*	Kan ^r^, DH5α containing pK18_∆*gacS*	This study
DH5α/pK18_∆*gacSA*	Kan ^r^, DH5α containing pK18_∆*gacSA*	This study
DH5α/pK18_∆*DJ41-1407*	Kan ^r^, DH5α containing pK18_∆*DJ41-1407*	[8]
DH5α/pK18_∆*gacADJ41-1407*	Kan ^r^, DH5α containing pK18_∆*gacADJ41-1407*	This study
S17-1λπ	*thi-1*, *thr*, *leu*, *tonA*, *lacY*, *supE*, *recA*, *RP4-2* (*Km::Tn7,Tc::Mu-1*), *Smr*, *lpir*	[8]
S17-1λπ/pK18_∆*DJ41_1407*	Kan ^r^, S17-1λπ containing pK18_∆*DJ41-1407*	[8]
S17-1λπ/pK18_∆*gacA*	Kan ^r^, S17-1λπ containing pK18_∆*gacA*	This study
S17-1λπ/pK18_∆*gacS*	Kan ^r^, S17-1λπ containing pK18_∆*gacS*	This study
S17-1λπ/pK18_∆*gacSA*	Kan ^r^, S17-1λπ containing pK18_∆*gacSA*	This study
S17-1λπ/pK18_∆*gacSDJ41_1407*	Kan ^r^, S17-1λπ containing pK18_∆*gacSDJ41-1407*	This study
S17-1λπ/pK18_∆*gacADJ41_1407*	Kan ^r^, S17-1λπ containing pK18_∆*gacADJ41-1407*	This study
*A. baumannii*		
ATCC 19606	Amp ^r^, clinical isolate, wild type	[43]
∆*DJ41_1407*	Amp ^r^, deletion of *DJ41_1407*	[8]
∆*gacS*	Amp ^r^, deletion of *gacS*	This study
∆*gacA*	Amp ^r^, deletion of *gacA*	This study
∆*gacSA*	Amp ^r^, deletion of *gacSA*	This study
∆*gacSDJ41_1407*	Amp ^r^, deletion of *gacSDJ41_1407*	This study
∆*gacADJ41_1407*	Amp ^r^, deletion of *gacADJ41_1407*	This study

Amp: Ampicillin, Kan: kanamycin.; the superscript “^r^” behind the antibiotic abbreviation denotes resistance to said antibiotic.

**Table 3 ijms-26-10620-t003:** Plasmids used in this study.

Plasmids	Description	References or Source
pK18mobsacB	Kan ^r^, mobilisable suicide vector, *sac*B, *ori*T	[35]
pK18_∆*DJ41_1407*	Kan ^r^, pK18*mobsacB* containing *DJ41_1407* upstream and downstream 1 kb fragments	[8]
pK18_∆*gacS*	Kan ^r^, pK18*mobsacB* containing *gacS* upstream and downstream 1 kb fragments	This study
pK18_∆*gacA*	Kan ^r^, pK18*mobsacB* containing *gacA* upstream and downstream 1 kb fragments	This study

Kan: Kanamycin.; the superscript “^r^” behind the antibiotic abbreviation denotes resistance to said antibiotic.

**Table 4 ijms-26-10620-t004:** Primers used in this study.

Primers	Sequence (5′-3′)	Application	References or Source
pK18-gacSUP_F	ATTCGAGCTCGGTACCCGGGCCATACTGCGACCTGAAAGC	Construct and confirm the *gacS* mutant	This study
pK18-gacSDO_R	TAAAACGACGGCCAGTGCCATTCTGCAATTTTTACTGAAG	Construct and confirm the *gacS* mutant	This study
gacSUP-gasSDO_F	GCCTATTTAACAACTATTATAGCTACATCAGATTGATCTC	Construct a *gacS* mutant	This study
gacSDO-gacSUP_R	GAGATCAATCTGATGTAGCTATAATAGTTGTTAAATAGGC	Construct a *gacS* mutant	This study
pk18-gacAUP_F	ACGACGGCCAGTGCCAGTGGTTGAGAACTGACGAAT	Construct and confirm the *gacA* mutant	This study
pk18-gacADO_R	GAGCTCGGTACCCGGGAGCGAGAGGGTTGCGGATCT	Construct and confirm the *gacA* mutant	This study
gacAUP-gacADO_F	GAAATTGCATGTAAGTTTGGGAAGACCTCCTTTCTTTCAA	Construct a *gacA* mutant	This study
gacADO-gacAUP_R	TTGAAAGAAAGGAGGTCTTCCCAAACTTACATGCAATTTC	Construct a *gacA* mutant	This study
pK18-A1S_0235_up_F	GAGCTCGGTACCCGGGACGGGTTATTGACGAGTTCT	Construct and confirm the *DJ41_1407* mutant	This study
A1S_0235_up_A1S_0235_do_R	GGTTGCTGTTCTGCCATTTCACGAAAAATCGTATGGGACA	Construct *DJ41_1407* mutant	This study
A1S_0235_up_A1S_0235_do_F	TGTCCCATACGATTTTTCGTGAAATGGCAGAACAGCAACC	Construct *DJ41_1407* mutant	This study
pK18-A1S_0235_do_R	ACGACGGCCAGTGCCAACTCCATACAAATTTTCTGA	Construct and confirm the *DJ41_1407* mutant	This study
gacA/07-IGR-qF	GCGATTCGTTACGGTTTGAT	Amplify intergenic region	This study
gacA/07-IGR-qR	AGGAATATAAGGCAGGTTGCTG	Amplify intergenic region	This study
*paa*_qF-2	AAGCAACAGGTGGCCGTGAT	qRT-PCR for *paa*	This study
*paa*_qR-2	ACCGACTTCACCTTCAACATACGC	qRT-PCR for *paa*	This study
Adh4-qF	TGCAAGATGAAGGGCTATTT	qRT-PCR for Adh4	[34]
Adh4-qR	CACCGCCTAACGACACAATA	qRT-PCR for Adh4	[34]
IPDC-F	ATATTGCTCAACCGCTTTGG	qRT-PCR for *ipdC*	This study
IPDC-R	GCAGCGTTTTCACCCATAAT	qRT-PCR for *ipdC*	This study
iaaHQ-F	GGTGGCTCTTCAAGTGGTTC	qRT-PCR for *iaaH*	This study
iaaHQ-R	AATTTGCAAAAGGACCAACG	qRT-PCR for *iaaH*	This study

## Data Availability

Data is contained within the article and Appendix A.

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
