# Peer review of "Roles of GacSA and DJ41_1407 in Acinetobacter baumannii ATCC 19606"

_ijms, 2025, doi:10.3390/ijms262110620_

Round 1

Reviewer 1 Report

Comments and Suggestions for Authors

Acinetobacter baumannii can be an opportunistic pathogen in humans, affecting people with compromised immune systems, and is becoming increasingly important as a hospital-derived (nosocomial) infection. Authors investigated roles of GacSA and DJ41_1407 in Acinetobacter baumannii ATCC 19606. In this manuscript, they find that DJ41_1407 phosphorylates GacA, all three proteins play an important role in the carbon metabolisms, colonial formation, virulence, antibiotic resistance. While several majors and minors should be taken into consideration seriously to improve this manuscript. And the language in this manuscript should also be improved.

Majors

  1. Transcriptomic short sequencing reads obtained in this study should be deposited into public database and assessable by others. Furthermore, transcriptomic sequencing should be performed by three replicates. And why were the mutant strains with ΔDJ41_1407 excluded from transcriptomic study? And I think those data are important in this study.
  2. The qRT-PCR results should be added in the abstract. Furthermore, those phenotypic determination results including the colony size, biofilm-forming ability, virulence, and antibiotic resistance should also be shown in the abstract.
  3. Which mutant strain is mentioned in line 59?
  4. Smears appear in the gel graph (Figure 1). Please provide a graph without obvious smears.
  5. Authors mentioned that several genes were expressed differently, while the analysis method determining those expression differences were absent in this manuscript. Please add it.
  6. Upregulated and downregulated genes should be described in detail in the transcriptomic results. And the expression pattern of the gacA and DJ41_1407 should also be described.
  7. How did authors identify the functions of DJ41_1407? Blast analysis againt RefSeq database? KEGG or GO annotations?
  8. How did authors conclude that the level of GacA phosphorylation by DJ41_1407 was lower compared to those of GacS on GacA?
  9. Expression differences should be tested by using the statistical method, such as the false discovery rate.
  10. Why was the ipdC expression upregulated by deleting DJ41_1407?
  11. The Figure 6C is suggested to be shown in a table.
  12. Authors mentioned that the colony sizes of the ΔgacA, ΔgacSA mutants were significantly smaller than those of other strains (lines 265-266, Figure 7A), while they also wrote that no significant differences were observed by using statistic analysis based the results of ImageJ (lines 270-271, Figure 7B). Please explain it. Moreover, I think that GacA has a weak effect on the colony size of Acinetobacter baumannii ATCC 19606.
  13. Please provide the manufacturers of LB medium and agar, antibiotics.
  14. Please add Table 4 in this manuscript.
  15. Are any transcriptomic differences affected by GacSA in the comparisons between Acinetobacter baumannii ATCC 19606 and A. baumannii ATCC 17198? If yes, please discuss it.
  16. Are DJ41_1407 homologs commonly present in the genomes of Acinetobacter baumannii, even other bacteria? I suggest authors to perform a comprehensively comparative genomic analysis to address this problem, which might highlight scientific significances of this study.
  17. Please merge 4.1 and 4.2 into a subsection.
  18. How did authors confirm that target genes were deleted in the mutant strains? The PCR gels confirming the presence and absence of those genes in the wild-type and mutant strains should be provided.

Minors

  1. Please change “ATCC19606” to “ATCC 19606” (line 3 and elsewhere).
  2. Please change “regulate” to “regulated” (line 17).
  3. Please change “gram-negative” to “Gram-stain-negative” (line 31 and elsewhere).
  4. Please provide the full name of ESKAPE (line 36).
  5. Please change “was” to “is” (line 46).
  6. Please change “exhibited” to “exhibits” (line 59).
  7. Please merge those two paragraphs into one (lines 61-74).
  8. Please change “GacSA regulates” to “GacA regulates” (line 307).
  9. Please change “ΔD gacA J41-1407” to “ΔgacA DJ41-1407” (Table 1).
  10. Please change “A. baumannii” to “A. baumannii ATCC 19606” (line 395 and elsewhere).
  11. Please give the full name of OD (line 404).
  12. Please add the space between the value and the degrees Celsius (line 418 and elsewhere).
  13. Table 3 should be moved following Table 2.
Comments on the Quality of English Language

English language in this manuscript should be improved.

Author Response

We thank the reviewer for providing many helpful suggestions and comments, and please find our response in the attached PDF.

Reviewer 2 Report

Comments and Suggestions for Authors

1. The introduction section is short and it would be helpful to expand it by including more relevant literature on the topic.
2. Were the transcriptomic data collected from biological replicates? Could you please provide information on the number of replicates used?
3. Do you plan to perform a proteomic analysis in order to validate the data obtained?
4. How precisely do you think GasA affects the physiology of the strain (colony size)?
5. Why were the antibiotics chosen in the concentrations mentioned? If there are any regulations regarding this, please provide a reference.
6. The discussion section could benefit from being improved: some information may be more appropriate for the introduction, and the results could be discussed more thoroughly and clearly.
7. Please add links to any applications or services that you used.
8. Please provide the composition of the culture media used in your study.

Author Response

(The authors gave the same response as above.)

Round 2

Reviewer 1 Report

Comments and Suggestions for Authors

Authors have revised this manuscript well, while two points are still needed to be considered before it can be accepted.

  1. Figure 1 can be moved to supplementary figure, due to several smears occurring.
  2. Bioinformatic tools and statistic analysis in the transcriptome investigation should be added in the subsection of 4.2, which should be entitled “Transcriptome sequencing and analysis”.

Author Response

We thank the reviewer for these helpful comments, and please see the attached PDF file for the point-by-point response.

Reviewer 2 Report

Comments and Suggestions for Authors

I consider the manuscript has been sufficiently improved.

Author Response

We thank the reviewer for the many helpful suggestions and comments that have helped to improve this manuscript, and please see the attached PDF file for the point-by-point response.
